# Antibodies as Snakebite Antivenoms: Past and Future

**DOI:** 10.3390/toxins14090606

**Published:** 2022-09-01

**Authors:** Wilmar Dias da Silva, Sonia A. De Andrade, Ângela Alice Amadeu Megale, Daniel Alexandre De Souza, Osvaldo Augusto Sant’Anna, Fábio Carlos Magnoli, Felipe Raimondi Guidolin, Kemily Stephanie Godoi, Lucas Yuri Saladini, Patrick Jack Spencer, Fernanda Calheta Vieira Portaro

**Affiliations:** 1Immuchemistry Laboratory, Butantan Institute, São Paulo 05503-900, Brazil; 2Biopharmaceuticals Laboratory, Butantan Institute, São Paulo 05503-900, Brazil; 3Laboratory of Structure and Function of Biomolecules, Butantan Institute, São Paulo 05503-900, Brazil; 4Nuclear and Energy Research Institute, São Paulo 05508-000, Brazil

**Keywords:** antivenom, venom, antibodies, snake bites, antivenom design, neglected tropical disease

## Abstract

Snakebite envenomation is considered a neglected tropical disease, affecting tens of thousands of people each year. The recommended treatment is the use of antivenom, which is composed of immunoglobulins or immunoglobulin fragments obtained from the plasma of animals hyperimmunized with one (monospecific) or several (polyspecific) venoms. In this review, the efforts made in the improvement of the already available antivenoms and the development of new antivenoms, focusing on snakes of medical importance from sub-Saharan Africa and Latin America, are described. Some antivenoms currently used are composed of whole IgGs, whereas others use F(ab’)2 fragments. The classic methods of attaining snake antivenoms are presented, in addition to new strategies to improve their effectiveness. Punctual changes in immunization protocols, in addition to the use of cross-reactivity between venoms from different snakes for the manufacture of more potent and widely used antivenoms, are presented. It is known that venoms are a complex mixture of components; however, advances in the field of antivenoms have shown that there are key toxins that, if effectively blocked, are capable of reversing the condition of in vivo envenomation. These studies provide an opportunity for the use of monoclonal antibodies in the development of new-generation antivenoms. Thus, monoclonal antibodies and their fragments are described as a possible alternative for the production of antivenoms, regardless of the venom. This review also highlights the challenges associated with their development.

## 1. Snakebites and Serum Therapy

The estimated number of snakebites in the world is about 400,000/year with approximately 20,000 deaths/year [1]. Another estimate shows that there are around 1.8–2.7 million snake envenomations annually, resulting in around 81,000–138,000 deaths, and that there may be as many as 400,000 people with permanent disabilities resulting from snakebite each year [2,3,4].

The greatest burden of snakebite envenomation occurs in Asia, sub-Saharan Africa, Latin America, and parts of Oceania [1]. In 2017, the World Health Organization considered ophidism a neglected tropical disease of the highest priority, and set a commitment with several countries to reduce snakebite accidents by 50% by 2030 [5].

In the Middle East and North Africa, 17 snake species are found, and, in sub-Saharan Africa, encompassing the Central, East, South, and West regions of the African continent, 26 species are found. Among these, the most medically relevant snakes belong to the genera *Echis* spp., *Naja* spp., *Dendroaspis* spp., and *Bitis* spp. (Figure 1) [6]. In Latin America, the most important snakes belong to the genera *Bothrops* spp., *Lachesis* spp., *Crotalus* spp., and *Micrurus* spp. [7,8].

The *Bitis* genus includes six species responsible for a large number of bites: *B. arietans*, *B. somalica*, *B. parviocula*, *B. gabonica*, *B. rhinoceros*, and *B. nasicornis* [9,10]. 

*B. arietans*, a snake of great medical importance, is responsible for a large number of serious accidents, mainly in children and rural workers in Africa [10,11]. The *Bitis arietans* bite causes local damage, such as necrosis, and systemic symptoms, such as fever, neutrophilic leukocytosis, thrombocytopenia, hemolysis, and bleeding, which can result in anemia, reduced resistance to infections, diffuse hemorrhage, myocardial damage, coagulopathy, hypotension, and death [9,10].

Regarding the genus *Dendroaspis*, *D. polylepis* stands out and is popularly known as black mamba, whereas *D. angusticeps* and *D. viridis* are called green mambas. The envenomation may cause hypotension, tachycardia, paresthesia in superior and inferior limbs, and respiratory failure in their victims [12]. Proteomics studies reveled that the *D. polylepis* venom is mainly composed of Kunitz-type molecules, which include mamba dendrotoxins (63%), three-finger toxins (31%), and metalloproteases (3%) [13].

The genus *Naja* is composed of a large number of snakes, totaling 33 species [14], including *N. naja*, *N. annulifera*, *N. melanoleuca*, *N. mossambica*, and *N. nigricollis*. Most species have neurotoxic venom, acting on the nervous system, causing paralysis. In addition, many venoms from *Naja* spp. have cytotoxic characteristics that cause swelling and necrosis, in addition to cardiotoxic components [15].

According to reports, cases of human casualties resulting from *E. ocellatus* envenoming are higher than those of all other African snakes combined, accounting for 90% of bites and over 60% of mortalities, and several thousand permanent disabilities [16,17].

In Latin America, the prominent genera are *Bothrops* spp. and *Crotalus* spp., as they account for more than 95% of reported accidents. *Bothrops* species are abundant [18], with a wide geographic distribution, since they have successfully colonized most of the South American territory [19,20]. The *Bothrops* genus is responsible for the majority of accidents in Brazil (around 85%). The accidents gain attention because of the gravity of the symptoms resulting from the complex mechanism of action of many toxins, such as Snake Venom Metallo Proteases (SVMPs), Snake Venom Serine Proteases (SVSPs) and phospholipases A2 (PLA2s) [21,22,23]. Accidents caused by the *Bothrops* genus result in symptoms characterized by hemorrhage, inflammation, and disturbances in the coagulation cascade [24], with local and systemic clinical manifestations. The local manifestations are characterized by edema, ecchymosis, pain, and blisters with serous, hemorrhagic, or necrotic content that may occur. In systemic manifestations, gingivorrhagia, microscopic hematuria, purpura, bleeding in recent wounds, intense hemorrhage, shock, and renal failure are observed in the most severe cases [24,25,26,27].

Other snakes of medical importance in Brazil belong to the genus *Crotalus* (i.e., *C. durissus terrificus*, *C. d. cascavela*, *C. d. collineatus*, *C. d. ruruima*, *C. vergandis*, *C. d. ruruima*, *C. d. marajoensis*). Usually, their venoms are mainly composed of neurotoxins and myotoxins, such as crotoxin [27,28], crotamin [29,30], and gyroxin [28,30]. Responsible for both the neurotoxic and systemic myotoxic effects characteristic of this venom, crotoxin comprises two sub-units that are non-covalently linked: the non-catalytic A (CA), or crotapotin, and the catalytic unit, crotoxin B (CB) and phospholipase A_2_ (PLA_2_) [31]. Crotapotin is an acidic polypeptide with no detectable enzymatic activity. PLA_2_ catalyzes the hydrolysis of the sn-2-acyl chain of phospholipids resulting in fatty acids and lysophospholipids. Crotapotin, working as a chaperon, potentiates the toxicity of PLA_2_ by about 35-fold [25,32,33]. It is important to note that snakes of the *Crotalus* genus, commonly called rattlesnakes, are the cause of a large number of accidents in North America, especially in the United States. Fortunately, accidents rarely cause morbidity or fatalities [31].

Thus, snake venoms are a complex mixture, composed mainly of proteins (±90–95%), in addition to peptides, carbohydrates, segments derived from nucleic acids, metal ions, biogenic amines, lipids, and free amino acids, which have different biological activities [32]. Effective antivenoms are expected to be able to neutralize the major toxins in a venom.

The only treatment for snakebite is the use of antivenoms, or serum therapy, which, when injected into an envenomed human bitten by a snake, mainly neutralizes the toxins of the venom used in its production. Snake antivenoms are specific immunoglobulins produced by fractionation of plasma generally obtained from large domestic animals, mainly horses, hyperimmunized with the venoms of interest. 

Specific neutralizing antibodies were first identified in the serum of experimental animals, i.e., rabbits or guinea pigs, immunized against diphtheria or tetanus toxins [34]. In 1894, the serum was successfully used for treating children suffering from severe diphtheria, and was manufactured by Burroughs Welcome, United Kingdom. In the same year, two groups, simultaneously but independently, described the antitoxic properties of the serum from rabbits and guinea pigs immunized against cobra and viper venoms [34,35]. In 1895, a similar procedure was repeated with horses immunized with snake venoms in India by Haffkine and in Vietnam by Lépnay [36]. The snake antivenom specificity—whose range includes the biting species—was demonstrated by Vital Brazil [37,38].

Antivenoms can be raised against the venom of a single species (monospecific) or against multiple species (polyspecific), and have already proven to be effective in preventing many of the lethal and damaging effects of envenomation [4]. Antivenom administration can reverse the major effects of envenomation, such as anti-hemostasis, neurotoxicity, and hypotension [4]. In contrast, venom-induced local damage is not well prevented by antivenoms unless it is administered soon after the bite [39]. In addition, despite being therapeutically efficient, some antivenoms currently used can induce adverse reactions.

The vast majority of antivenom manufacturers refine IgG extracted from animal plasma, producing F(ab’)2 fragments through enzymatic digestion with pepsin. Other industries use papain to produce even smaller Fab fragments, improving safety and tissue distribution. This IgG format, however, has the disadvantage of rapid renal clearance. Removal of the Fc fraction of the antigen-binding fragment reduces the risk of adverse reactions [40]. Nonetheless, some antivenoms are made by whole IgG molecules, which are purified from plasma by caprylic acid precipitation or affinity chromatography, which yield a safer product but at a higher cost [4].

## 2. Immunoglobulins for Therapeutic and Diagnostic Applications 

The most widely used immunoglobulin for therapeutic and diagnostic applications [41], human IgG (Figure 2), is a protein made up of two identical heavy chains and two identical light chains, κ or λ, that are interconnected by a series of disulfide bonds. Each heavy chain contains three constant domains (i.e., CH1, CH2, and CH3) and one variable domain (VH). The light chain contains one constant domain (CL) and one variable domain (VH). The variable regions, at the N-terminus of the antigen-binding fragment (Fab), determine the specificity, diversity, and affinity of antigen binding. Within each variable domain, there are three hypervariable regions called complementary determining regions (CDR1, CDR2, and CDR3), which are primarily responsible for antigen recognition and binding. The remainder of the VH and VL domains are the framework regions that act as a scaffold to support the CDR’s loops. The CDRs and framework regions in each of the variable domains contribute to antigen binding [41]. 

Regarding formats, IgGs are the most abundant class of monoclonal antibodies (mAbs) approved for therapeutic use, accounting for 82% of the total [42]. FDA-approved antibodies can be classified as humanized (46%), human (41%), chimeric (10%), or murine (3%) [42], and are powerful tools for use in therapy and diagnosis. mAbs are one of the fastest growing classes of therapeutic biomolecules, with a market valued at USD 168.70 billion in 2021, and projected to reach USD 188.18 billion in 2022, recording a growth rate of 11.5% [43]. 

The first mAb approved for human use by the FDA was muromonab-CD3 (Orthoclone OKT3) in 1986, a murine mAb that acts as an immunosuppressant for acute transplant rejection, targeting CD3 expressed by T cells [42]. Currently, more than 100 monoclonal antibodies are approved by the FDA, with 6–12 approvals per year [44]. These numbers are expected to increase as technological advances have made research and development of monoclonal antibodies cheaper, faster, and more efficient.

Antibody engineering has also focused its attention on the development of functional antibody fragments, which have more favorable features, such as a smaller molecular size and higher affinity, when compared with whole IgGs (Figure 2) [45]. Currently, antibody fragments represent 9% of the total number of antibodies approved for therapeutic or diagnostic purposes [44]. Several antibody fragments are already in clinical trials, with antigen-binding fragments and single-chain variable fragments (scFvs) representing the majority [46].

There has also been a growing interest in heavy-chain only antibodies (HcAbs), the smallest naturally occurring antigen-binding fragment formed by a single heavy chain with a variable domain found in camelids or sharks (Figure 2). As the smallest known functional antibody fragments, nanobodies (VhHs) are molecules of approximately 15 kDa. Their small size, high solubility, tissue permeability, and stability to changes in pH and temperature, allow interaction with sites inaccessible to conventional antibodies and having low immunogenicity. Thus, nanobodies are promising candidates for different applications in biomedical research, diagnosis, and therapy [47]. 

Unlike Fab or (Fab’)2, an scFv is not a fragment of an antibody, but rather a protein formed by the fusion of VH and VL variable regions, connected by a short linker (Figure 2). This linker is rich in glycine, to provide flexibility, in addition to serine or threonine, to increase solubility and to also allow the connection from the N-terminus of the VL to the C-terminus of the VH, or vice versa [48]. This fusion protein retains the specificity of the original IgGs despite the removal of constant regions and the linker introduction.

ScFvs have some advantages when compared to chimeric or humanized antibodies, since these molecules maintain binding specificity and, due to their size and absence of a constant region, have little immunogenicity, better tissue penetration, and do not activate the complement system. Furthermore, scFvs can be produced in a prokaryotic expression system, as these molecules do not require glycosylation. However, their binding affinities and half-lives can be reduced [48]. As of June 2022, there were 4 Fabs, 2 scFvs, and only one humanized nanobody approved by the FDA for use [49].

Essentially, there are two ways in which antibodies can neutralize toxins, either by directly or indirectly inhibiting them. In direct inhibition, the antibody binds to the toxin and competes for the site of interaction, whether of an enzymatic toxin or not [50]. Indirect inhibition can occur through three different mechanisms: (i) allosteric inhibition, where the antibody binding to the toxin induces a conformational change, losing activity [51]; (ii) through a steric hindrance effect where there is binding of the antibody to a region close to the active site; or (iii) by preventing the dissociation of toxin complexes, preventing the formation of active toxins [52]. Considering the remarkable progress in the mechanisms underlying the immune response, and the immunochemistry of immunoglobulins and the high number of snakebites, studies have been designed with the objective of improving the neutralizing properties of antibody-antitoxins.

## 3. Classic Methods for Improving Serum Therapy

Currently, about 31 antivenoms are available worldwide for the treatment of snakebites, as listed by Laustsen et al. [53]. Despite their importance for the current treatment envenomation, conventional antivenoms are partially ineffective in neutralizing some effects caused by snake venoms in vitro and in vivo [54,55], and they can also cause some side effects [56,57]. In in vivo studies, the neutralization potential of the anti-bothropic serum, either with the treatment of the animals before or immediately after the application of the bothropic venom, was studied by Battellino et al. (2003), where different application schedules of the anti-bothropic serum produced by the Butantan Institute were analyzed: before (15 min), at the same time, and after (15 min) the application of *B. jararaca* venom [58]. The authors concluded that the low neutralization did not occur due to the lack of specific antibodies, but due to the low interaction of the antibodies with the venom components, probably due to the difficulty in distributing the F(ab’)2 fragments in the tissues. It is important to emphasize that the results indicate the formation of hemorrhagic lesions even with the in vitro pre-incubation procedure, but with a proportion almost six times lower when compared to the control. Blind studies with humans envenomed by the snake *Echis ocellatus* in Ghana showed that a new antivenom was ineffective in combating lethality, demonstrating the need for pilot tests before its single and general distribution in a region is initiated [54].

For this reason, new strategies have been used to optimize the treatment of snakebites, whether designing experiments to obtain new high-quality antivenoms or improving the effectiveness of current commercial antivenoms. 

Some strategies for developing antivenoms are based on cross-reactivity. The evaluation of the cross-reactivity of antivenoms, i.e., the ability of antibodies to neutralize toxins from different snake species, yields the possibility of attaining paraspecific antivenoms, which can be easier to produce and more profitable [59]. Some studies show that, although antivenoms are produced from a limited number of species, they show very interesting cross-reactivity with other species from the same region. Among several examples [60,61], the cross-reactivity between nine snake venoms of the genus *Bothrops* and a serum produced against the venom of *B. jararaca* stands out [62]. In fact, despite the diversity of this genus in Brazil, the victims are treated with the pentavalent serum produced by the Butantan Institute, which is attained using the venoms of *B. jararaca* (50%), *B. jararacussu* (12.5%), *B. alternatus* (12.5%), *B. moojeni* (12.5%), and *B. neuwiedi* (12.5%) [24,62] as antigens. Moreover, another study aiming at the development of antivenoms against eight snake species found in Mozambique resulted in high titers of Abs against *Bitis arietans*, *B. nasicornis,* and *B. rhinoceros* (5.18 × 10^6^, 3.60 × 10^6^, and 3.50 × 10^6^ U-E/mL, respectively), and against *Naja melanoleuca*, *N. mossambica*, and *N. annulifera* (7.41 × 10^6^, 3.07 × 10^6^, and 2.60 × 10^6^ U-E/mL, respectively), but lower titers against the *Dendroaspis angusticeps* and *D. polylepis* venoms (1.87 × 10^6^ and 1.67 × 10^6^ U-E/m, respectively) [63].

An SVMP with hemorrhagic activity present in the venom of *B. arietans* was recognized by antibodies present in a series of polyvalent antivenoms, composed of F(ab)’2 portions, attained by horse and chicken immunizations. In addition to anti-*Bitis* spp. serum, anti-*Bothrops* spp., anti-*Lachesis muta*, anti-*Crotalus* spp., and anti-*Naja* spp. recognized the purified SVMPs. The observed cross-reactivity indicates that metalloproteases induce an immunological signature, probably due to the presence of common epitopes among the different SVMPs present in various snake venoms [64]. 

The use of adjuvants can also improve antivenom antibody titers. Using the venom of *Crotalus d. terrificus* as an antigen and different immunization protocols, it was shown that venom emulsified with Freund’s adjuvant induced a more protective and sustained immune response compared to Al(OH)_3_ or liposome particles [65]. In addition to adjuvants, chicken hyperimmunization has also demonstrated the effectiveness of IgYs in recognizing, combining, and neutralizing the toxic and lethal components present in venoms from snakes of the *Bothrops* and *Crotalus* genera [66]. 

Another simple strategy that can improve the quality of an antivenom is changing the immunization protocol and the amount of immunogen injected. Increasing the interval between boosters and using a smaller amount of venom appear to result in antibodies with higher titers and affinity [67]. 

Several studies have indicated that, when purified toxins are used as antigens, instead of the total venom where they are present, they favor the production of antibodies with better titers and affinity [68]. In one study, the use of crude *Crotalus* venom, and purified crotoxin and PLA_2_ present in this venom, as immunogens in horses were compared. The results indicated that the serum obtained against the total venom showed both low titers and neutralizing capacity. By comparison, immunization with crotoxin resulted in a serum with greater specificity and, if used as an adjuvant, can prevent injuries and adverse reactions in serum-producing animals. Finally, immunization with PLA_2_ resulted in less neutralizing serum, especially when PLA_2_ epitopes are in their free form. In general, antibodies with high titers and the ability to cross-react were produced, but there was no increase in affinity [69]. Using purified toxins, a potential new antivenom was developed against SVSPs from *Bothrops jararaca* venom with enzymes identified as unblocked by the antivenom produced by the Butantan Institute. Isogenic C57BL/6 and BALB/c mice were immunized with a pool of four purified serine proteases (KN-BJ2, BjSP, HS112, and BPA), which were not inhibited by commercial antiserum. The results showed that the two sera obtained were able to block the SVSPs of five bothropic venoms, indicating that the use of purified toxins can further improve the quality of the sera [70,71].

## 4. Next-Generation Antivenom: Monoclonal Antibodies and Their Fragments

The discovery and development of monoclonal antibodies became cheaper, quicker, and more efficient in recent years, paving the way for the development of “next-generation antivenom”. This is the use of human monoclonal antibody mixtures that target the key toxins in snake venoms. Toxins from snake venoms feature a synergic effect, so their action provokes a greater toxification and/or lethality than the sum of their separate effects. As a consequence, neutralizing certain key toxins from the venom is enough to drastically reduce the effects of the whole envenomation [72,73]. 

Various different monoclonal antibodies have been discovered and developed against toxins from different venomous animals, such as snakes, scorpions, spiders, and bees, as reviewed by Lausten and colleagues [53]. 

The first stage in these developments occurred in 1982, when Boulain and colleagues developed the first homogeneous population of high-affinity monoclonal antibodies specifically for *Naja nigricollis* snake α-toxin [74]. The spleen cells were fused with myeloma to obtain a hybridoma, and the resulting toxin-binding antibodies were purified and tested. This antibody neutralizes the biological activity of the toxin under both in vivo and in vitro conditions [74]. In 1995, Meng and collaborators used, for the first time, the technique of phage display to find monoclonal antibody fragments against animal toxins. In this study, scFvs from the library of human semi-synthetic antibodies against crotoxin were identified [75], and two years later Lafaye and colleagues developed the first human scFv targeting venom toxins using phage display [52].

Another strategy gaining momentum is the use of humanized or human monoclonal antibodies. These antibodies, which are compatible with the human organism, present a lower risk of reducing the immune characteristics of the antivenom [76]. The first fully human IgG was developed against hemorrhagic metalloproteases from *Protobothrops flavoviridis*, and was attained from the fusion of myeloma cells SP2/0-Ag-14 and cells from KM mice spleens previously immunized with the toxin. In this study, 300 hybridoma cells were produced to attain IgGs for the toxin HR1a, and 80 reactive antibodies were identified [77].

In 2018, the first development of fully human monoclonal IgGs against snake venom using phage display was reported. The IgGs were selected from a naïve human library of scFvs, and showed specificity to dendrotoxins from *Dendroaspis polylepis*. Monoclonal antibodies were able to prevent lethality when pre-incubated with the toxin fraction and, in addition, showed that an oligoclonal mixture of toxin-binding antibodies was able to prevent lethality for the entire venom [53].

The firsts antibody formats for recombinant antivenom were monoclonal IgGs and single-chain variable fragments. In vivo studies have shown that monoclonal IgGs targeting snake toxins are able to neutralize myotoxic, hemorrhagic, and proteolytic effects. Monoclonal IgGs that neutralize these effects were developed for some medically relevant snakes from the genera *Naja* spp., *Crotalus* spp., *Echis* spp., *Laticauda* spp., and *Bothrops* spp. [50,74,78,79,80,81]. Monoclonal antibodies against a phospholipase A2, in addition to a metalloproteinase and a thrombin-like via hybridoma, were obtained and, when used together, prevented the in vivo lethality of *Bothrops atrox* venom [80]. Other studies have also been published on monoclonal IgGs targeting *Bothrops* spp. toxins, such as the development of a mAb that neutralizes the hemotoxic effects of Atroxlysin-I from *B. atrox* venom [82]. In 1988, Lomonte and Kahan developed the first mAb against *B. asper*, a murine IgG that neutralizes the myotoxic effects from the venom [83]. In 2010, mAbs capable of neutralizing *B. asper* BaP1 in the nanomolar range were developed, blocking its hemotoxic effects [84]. To our knowledge, no mAb-targeting *Bitis* spp. toxins have been developed.

Another antibody format widely used in snakebite immunotherapy is Fab, which is mainly used in conventional serum therapy, in a polyclonal mixture. Two Fabs targeting snake toxins, cardiotoxin from *Naja nigricollis* and b1-bungarotoxin from *Bungarus multicinctuse*, neutralize their in vitro and in vivo effects [85,86].

The single-chain variable fragment format has been extensively studied and developed for snakebite immunotherapy, and several studies have identified human scFvs capable of decreasing the effects on *Bothrops jararacussu* and *Crotalus durissus* toxins [75,87,88,89,90,91,92,93]. Due to their small size, although most IgGs are expressed in hybridoma cells or mammalian cells, different expression systems can be used to produce scFvs, with microbial systems being the most used, and even plants have been used [94,95]. 

The use of recombinant nanobodies that are able to recognize medically relevant snake venom toxins has also gained attention [96,97,98]. Using a naive VHH library, clones against α-cobrotoxin present in *Naja kaouthia* venom were identified [99]. Humanized-single domain antibodies for the phospholipase present in the *N. kaouthia* venom were able to inhibit this activity up to 50% [100]. Using a library of VHH immune genes, a clone with high affinity for α-cobratoxin was selected and fused to a human Fc fragment to create a VHH2-Fc antibody that, when pre-incubated with the toxin, neutralized its lethality [101]. An immune VHH library for phospholipases A2 from *Bothrops jararacussu* venom was constructed, and the selected clones neutralized in vivo myotoxic activity, and presented cross-reactivity with PLA2 from different *Bothrops* species [102]. Moreover, from an immune VHH library, clones against the hemorrhagic and myotoxic fractions of *B*. *atrox* venom were selected and were able to neutralize these effects; however, the nanobodies were not able to prevent lethality [98].

These studies show that monoclonal antibodies and/or antibody fragments, as an innovative antivenom for viperid and elapid species, are feasible, and may be the next step in snakebite therapy. Antibody fragments, due to their small size, diffuse rapidly through the body, reaching a higher tissue biodistribution when compared to conventional serum therapy [53]. 

Research regarding the improvement in heterologous antivenom is extremely important and, to this day, it is the only effective and available treatment for snakebite envenomation. In countries that suffer the highest rate of snakebites, the production of recombinant antivenom may seem unfeasible, since the research and development of monoclonal antibodies requires a high level of technology [42]. However, the high investment required can be offset by a cheaper final product compared to heterologous sera. Theoretically, an oligoclonal mixture of antibodies with 25% cross-reactivity can equal the costs of current treatment and, when taking into account different formats and even expression strategies, the treatment can be up to 10 times cheaper [103]. The technology needed for antibody research and development has become cheaper every year, with some estimates even surpassing Moore’s law [104].

Finally, the use of animals for research, development, and manufacturing has been a controversial topic for many years, and there is a public desire to reduce animal use, particularly where non-animal-derived alternatives are present [105]. Since the treatment of snakebites has historically used immunization of large mammals, the substitution by recombinant sera, made from a cocktail of monoclonal antibodies, may be a safer—and even cost-competitive—alternative for future therapy of envenomation [106,107].

## Figures and Tables

**Figure 1 toxins-14-00606-f001:**
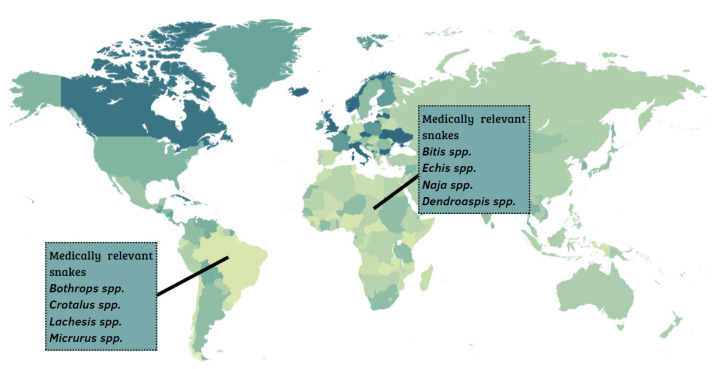
Geographic distribution of the most medically important snake species in Africa and Latin America.

**Figure 2 toxins-14-00606-f002:**
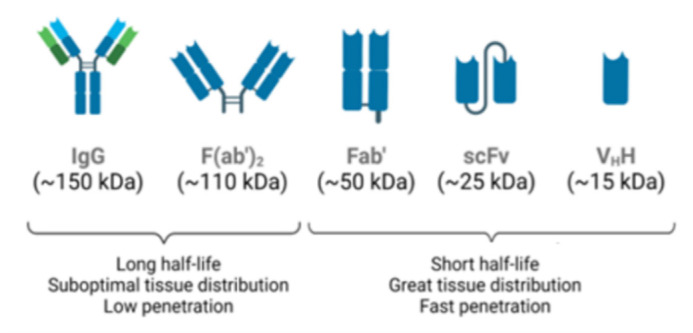
Different antibody formats with biological activities. IgG: whole molecule. F(ab’)2: product of IgG digestion with pepsin. Fab: product of digestion of IgG with papain. Diabody: non-covalent dimers of scFv fragments. scFv: single-chain variable fragments. VhH: single domain antigen specific fragment. Schematic diagram developed on the biorender platform.

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
