# Peer review of "Antibodies as Snakebite Antivenoms: Past and Future"

_toxins, 2022, doi:10.3390/toxins14090606_

Round 1

Reviewer 1 Report

Review article entitled “Antibodies as antivenoms: Past and future” is well written and summarized the useful and interesting knowledge about antivenoms against snake bite with principal papers including recent published ones.

This review article is acceptable for publication in Toxins, some minor points should be addressed;

Major:

This review article focused on the antivenoms against venomous snakes but no description antivenoms against other venomous animals. So, I recommend that the title of “Antibodies as antivenoms: Past and future” is slightly changed to more specific to the antivenoms against venomous snakes.

Minors:

References 9, 14, 16 and 23: More detailed informations such as website address should be described.

References 29, 35, 37 40 42, 44, 45, 46, 47, 57 and 63: Number of volumes and pages should be described.

Author Response

Reviewer 1

The authors traced the history of antibody therapy against snake venom, 1) Explanation of antibodies used in antibody therapy, 2) Snake bite wounds and antibody therapy, 3) Historical antibody production methods and treatment methods, 4) Next-generation antibody production technology It is very easy to understand because it is discussed separately. As a reviewer, I would like to discuss how to proceed in the future in these countries, as poisonous snake bites often occur in economically disadvantaged countries regarding next-generation antibody production, which requires a high level of technology and is expensive.

Answer:  We appreciate the comments and agree to add a discussion on the challenges for the production of new generation antivenoms in developing countries. For this we have added a new paragraph and two new references as indicated below:

Line 471: The study regarding the improvement of heterologous antivenom is extremely important and to this day, their use is the only effective, and available, treatment for snakebite envenomation. In countries that suffer the highest rate of snakebites the production of recombinant antivenom may seem unfeasible, since the research and development of monoclonal antibodies requires a high level of technology [40]. How-ever, the high investment required can be offset by a cheaper final product compared to heterologous sera. Theoretically, an oligoclonal mixture of antibodies with 25% cross-reactivity could equal the costs of current treatment, and when taking into account different formats and even expression strategies, the treatment could be up to 10 times cheaper [104]. The technology needed for antibody research and development has become cheaper every year, with some even surpassing Moore's law [105].

Reviewer 2 Report

The authors traced the history of antibody therapy against snake venom, 1) Explanation of antibodies used in antibody therapy, 2) Snake bite wounds and antibody therapy, 3) Historical antibody production methods and treatment methods, 4) Next-generation antibody production technology It is very easy to understand because it is discussed separately. As a reviewer, I would like to discuss how to proceed in the future in these countries, as poisonous snake bites often occur in economically disadvantaged countries regarding next-generation antibody production, which requires a high level of technology and is expensive.

Author Response

Reviewer 2

This review is of considerable interest to the scientific community involved in the study of Snakebite envenomation and to the community involved in the development of drugs based on recombinant antibodies.

The review is clear and comprehensive, so in my opinion it can be accepted in its present form.

Answer: Thanks so much for the comments.

Minor:

Page 4 Line 138: ‘PLA2s (phospholipases A2)’ should be written instead of ‘PLAs’

Answer: We agree.

Line 74: The accidents call attention because of the gravity of the symptoms due to the complex mechanism of action of many toxins, such as SVMPs (Snake Venom Metallo Proteases), SVSPs (Snake Venom Serine Proteases) and PLA2s (phospholipases A2) [21–23].

Reviewer 3 Report

This review is of considerable interest to the scientific community involved in the study of Snakebite envenomation and to the community involved in the development of drugs based on recombinant antibodies.

The review is clear and comprehensive, so in my opinion it can be accepted in its present form.

Minor:

Page 4 Line 138: ‘PLA2s (phospholipases A2)’ should be written instead of ‘PLAs’

Author Response

Reviewer 3

The authors present a review of antibody-based antivenoms, encompassing the history, technological development, and future of this therapy.

1- The first section of the work presents basic antibody development, antibody type and potential uses of antibody fragments. While I usually do not suggest this, the authors should place this section after the next 1-2 sections of the work.

Answer: We agree that changes in the order in which topics are presented make the text more understandable. Changes were made as suggested by the Reviewer

2- In order to better frame the issues at hand, it is better to discuss snake bite and antivenom first as is seen in section 2. The need for antivenom development was driven by the clinical disease burden caused by the snakes mentioned in section 2. I note that for some peculiar reason the snakes of North America are omitted. The burden caused by rattle snakes and other vipers should be presented.

Answer: We agree that accidents caused by rattlesnakes in North America are relevant in number and degree, and this fact should be considered in the review. Thus, a paragraph was added to address this topic in the review, as well as a new reference.

Line 91: It is important to inform that snakes of the genus Crotalus, commonly called ratllesnakes, are the cause of a large number of accidents in North America, especially in the United States. Fortunately, accidents rarely cause morbidity or fatalities, with around 1000 and four deaths each year [31].

[31] Walter, F.G.; Stolz, U.; Shirazi, F.; McNally, J. Epidemiology of Severe and Fatal Rattlesnake Bites Published in the American Association of Poison Control Centers’ Annual Reports. Clin Toxicol 2009, 47, 663–669, doi:10.1080/15563650903113701.

3- Section 3 goes into detail concerning specific antivenoms being developed for the various species. Given this, section 1 would be best placed after section 2 and before section 3. I would also strongly suggest that the authors create a table with these antivenoms to venoms/specific enzymes to better organize the information in a place easily accessed by the readership.

Answer: As described above, changes were made to the order in which topics were presented, and we agree that the text has improved with the suggested changes. We also agreed that a table summarizing the available antivenoms and their targets would be an important upgrade for the review. However, this table has already been provided by Laustsen et al., in the article Pros and cons of different therapeutic antibody formats for recombinant antivenom development (A.H. Laustsen et al. / Toxicon 146 (2018) 151e175152). Thus, we ask for the understanding of this Reviewer so that we do not add the requested table, but rather emphasize the existence of this data in the cited publication, as described below:

Line 304:

  1. Classic methods for improving serum therapy

Currently, about 31 antivenoms are available worldwide for the treatment of snakebites, as listed by Laustsen et al. [77]. Despite of their importance on the current treatment envenomation, conventional antivenoms are partially ineffective in neutralizing some effects caused by snake venoms in vitro and in vivo [48,49], and they can also cause some side effects [50,51].

4- It would be of interest if the authors could review what sites on the venom enzymes of interest are bound by the various antibodies. Are there regions of highly conserved amino acids that are where the antibodies bind? Are they in catalytic sites, or are they elsewhere, preventing steric interaction with venom enzyme substrates. This would add a molecular biology insight into the discussion.

Answer: We agree that the addition of already identified epitopes and paratopes could enrich the text, however, we believe that approaching this aspect in a more comprehensive way could be more interesting for the present Review. Therefore, we ask for the understanding of this Reviewer to add the mechanisms of inhibition of an antitoxin antibody, as described below:

Line 190: Essentially, there are 2 ways that antibodies can neutralize toxins, either by directly or indirectly inhibiting them. In direct inhibition, the antibody binds to the toxin and competes for the site of interaction, whether of an enzymatic toxin or not [69]. In the case of indirect inhibition, it can occur through three different mechanisms: (i) Allosteric inhibition, where the antibody binding to the toxin induces a conformational change, losing activity [70]; (ii) Through a steric hindrance effect where there is binding of the antibody to a region close to the active site or (iii) preventing the dissociation of toxin complexes, preventing the formation of active toxins [71].

5- The last 2 paragraphs of section 3 go off topic and briefly discuss PLA2 inhibitors and peptidic inhibitors. Frankly, the authors should either omit this information or create another section that encompasses these and other small molecular weight inhibitors that have been extensively published, such as ruthenium-based approaches that inhibit all three major enzyme groups of venom (SVMP, SVSP, PLA2).

Answer: We totally agree. The paragraphs were removed.

6- Section 4 is thematic but not particularly focused. Again, a table could help, but better yet, a forward-looking figure that incorporates the new antibodies would be of interest.

Answer: We agree. In an attempt to focus more on monoclonal antibodies and their fragments as snake antivenoms, we consider it important to include this information in the title of Section 4. Furthermore, we emphasize that mAbs and their fragments have already been developed and that the results are satisfactory. Thus, considering the reasoning mentioned above during the development of the work, we kindly ask for the understanding and permission of this Reviewer to maintain the current text of Section 4. 

Line 393:

  1. Next-generation antivenom: monoclonal antibodies and their fragments

Reviewer 4 Report

The authors present a review of antibody-based antivenoms, encompassing the history, technological development, and future of this therapy.

 The first section of the work presents basic antibody development, antibody type and potential uses of antibody fragments. While I usually do not suggest this, the authors should place this section after the next 1-2 sections of the work.

In order to better frame the issues at hand, it is better to discuss snake bite and antivenom first as is seen in section 2. The need for antivenom development was driven by the clinical disease burden caused by the snakes mentioned in section 2. I note that for some peculiar reason the snakes of North America are omitted. The burden caused by rattle snakes and other vipers should be presented.

Section 3 goes into detail concerning specific antivenoms being developed for the various species. Given this, section 1 would be best placed after section 2 and before section 3. I would also strongly suggest that the authors create a table with these antivenoms to venoms/specific enzymes to better organize the information in a place easily accessed by the readership.

It would be of interest if the authors could review what sites on the venom enzymes of interest are bound by the various antibodies. Are there regions of highly conserved amino acids that are where the antibodies bind? Are they in catalytic sites, or are they elsewhere, preventing steric interaction with venom enzyme substrates. This would add a molecular biology insight into the discussion.

The last 2 paragraphs of section 3 go off topic and briefly discuss PLA2 inhibitors and peptidic inhibitors. Frankly, the authors should either omit this information or create another section that encompasses these and other small molecular weight inhibitors that have been extensively published, such as ruthenium-based approaches that inhibit all three major enzyme groups of venom (SVMP, SVSP, PLA2).

Section 4 is thematic but not particularly focused. Again, a table could help, but better yet, a forward-looking figure that incorporates the new antibodies would be of interest.

Author Response

Reviewer 4

Review article entitled “Antibodies as antivenoms: Past and future” is well written and summarized the useful and interesting knowledge about antivenoms against snake bite with principal papers including recent published ones.

This review article is acceptable for publication in Toxins, some minor points should be addressed;

Major:

This review article focused on the antivenoms against venomous snakes but no description antivenoms against other venomous animals. So, I recommend that the title of “Antibodies as antivenoms: Past and future” is slightly changed to more specific to the antivenoms against venomous snakes.

Answer: We agree. A new title were provided.

Antibodies as snakebite antivenoms: Past and future

Minors:

References 9, 14, 16 and 23: More detailed informations such as website address should be described.

References 29, 35, 37 40 42, 44, 45, 46, 47, 57 and 63: Number of volumes and pages should be described.

Answer: Thanks for the comments. Information has been added as described bellow.

Former reference 9, current #49: The Antibody Society Antibody Therapeutics Approved or in Regulatory Review in the EU or US. Available Available online: https://www.antibodysociety.org/resources/approved-antibodies/ (accessed on 23 August 2022).

Former reference 14, current #5: WHO Snakebite: WHO Targets 50% Reduction in Deaths and Disabilities. Available online: https://www.who.int/news/item/06-05-2019-snakebite-who-targets-50-reduction-in-deaths-and-disabilities (accessed on 23 August 2022).

Former reference 16, current #7: WHO; World Health Organization Guidelines for the Production, Control and Regulation of Snake Antivenom Immunoglobulins; Geneva, Switzerland, 2016. (https://extranet.who.int/pqweb/vaccines-production-control-and-regulation-snke-antivenom-immunoglobulins)

Former reference 23, current #14: The Reptile Database. Available online: http://reptile-database.org/ (accessed on 23 August 2022).

Former reference 29, current #20: Melgarejo, A.R.; Cardoso, J.L.C.; França, F.O.S.; Wen, F.H.; Málaque, C.M.S.; Haddad JR. Serpentes peçonhentas do Brasil. In, V. (Ed). Animais Peçonhentos no Brasil – biologia, clínica e terapêutica dos acidentes. 2.ed. São Paulo: Sarvier, 2008. p.42-70.

Former reference 35, current #26: Azevedo-Marques, M.M.; Cupo, P.; Coimbra, T.M.; Hering, S.E.; Rossi, M.A.; Laure, C.J. Myonecrosis, Myoglobinuria and Acute Renal Failure Induced by South American Rattlesnake (Crotalus durissus Terrificus) Envenomation in Brazil. Toxicon 1985, 23, 631–636, doi:10.1016/0041-0101(85)90367-8.

Former reference 37, current #28: Barrabin, H; Martiarena, JL; Vidal, JC & Barrio, A. Isolation and characterization of gyroxin from Crotalus durissus Terrificus venom. Rosemberg, P, (ed), Toxins Animal, Plant and Microbial. 1978 pp. 113-133, Pergamon Press, Oxford.

Former reference 40, current #33: Behring, E. von Ueber Das Zustandekommen Der Diphtherie-Immunität Und Der Tetanus-Immunität Bei Thieren. Molecular Immunology 1991. 28, issue 12, 1319-1320

Former 42, current #35: Bochner, R. and Goyffon, M. L'oeuvre Scientifique De Césaire Phisalix (1852-1906) Découvreur Du Sérum Antivenimeux. Bulletin De La Société Herpétologique De France, p.32. vol. 123. 2007.

Former 44, current #37: Brazil, V. Memória Histórica do Instituto Butantan. São Paulo, Elvino Pocai, 1941, 105-112.

Former 45, current #38: Brazil V.; Maibon J.. 1914. La Défense Contre L'ophidisme. 2. Éd. ed. Saint-Paul: Impr. Pocai-Weiss.

Former 46, current 39: Cardoso, J.L.; Fan, H.W.; França, F.O.; Jorge, M.T.; Leite, R.P.; Nishioka, S.A.; Avila, A.; Sano-Martins, I.S.; Tomy, S.C.; Santoro, M.L. et al. Randomized comparative trial of three antivenoms in the treatment of envenoming by lance-headed vipers (Bothrops jararaca) in São Paulo, Brazil. Q J Med. 1993 May;86(5):315-25. PMID: 8327649.

Former 47, current 40: Kang, T.H.; Jung, S.T. Boosting therapeutic potency of antibodies by taming Fc domain functions. Exp Mol Med 51, 1–9 (2019). https://doi.org/10.1038/s12276-019-0345-9.

Former reference 57, current #63: Dias da Silva, W.; Guidolin, R.; Raw, I.; Higashi, H.G.; Caricati, C.P.; Morais, J.F. et al. Cross-reactivity of horse monovalent antivenoms to venoms of ten Bothrops species. Mem Inst Butantan. 1989;51: 153–168.

Former reference 63, current #69: Guidolin R. G., Marcelino R. M., H. H. Gondo, J. F. Morais, R. A. Ferreira, C. L. Silva, T. L. Kipnis, J. A. Silva, J. Fafetine and W. D. da Silva. Polyvalent horse F(Ab`)2 snake antivenom: Development of process to produce polyvalent horse F(Ab`)2 antibodies anti-african snake venom. African Journal of Biotechnology 9, no. 16 (2010): 2446-2455..

Round 2

Reviewer 4 Report

No further comments. Well done.